# Potential Effects of Romanian Propolis Extracts against Pathogen Strains

**DOI:** 10.3390/ijerph19052640

**Published:** 2022-02-24

**Authors:** Mihaela Laura Vică, Mirel Glevitzky, Ramona Cristina Heghedűş-Mîndru, Ioana Glevitzky, Horea Vladi Matei, Stefana Balici, Maria Popa, Cosmin Adrian Teodoru

**Affiliations:** 1Department of Cellular and Molecular Biology, “Iuliu Hațieganu” University of Medicine and Pharmacy, 400012 Cluj-Napoca, Romania; mvica@umfcluj.ro (M.L.V.); horea_matei@yahoo.com (H.V.M.); sbalici@umfcluj.ro (S.B.); 2Faculty of Exact Science and Engineering, “1 Decembrie 1918” University of Alba Iulia, 510009 Alba Iulia, Romania; mirel_glevitzky@yahoo.com (M.G.); mariapopa2010@yahoo.com (M.P.); 3Faculty of Food Processing Technology Banat’s, University of Agricultural Sciences and Veterinary Medicine, 300645 Timișoara, Romania; ramo75ro@yahoo.com; 4Doctoral School, Faculty of Engineering, “Lucian Blaga” University of Sibiu, 550025 Sibiu, Romania; 5Clinical Surgical Department, Faculty of Medicine, “Lucian Blaga” University of Sibiu, 550025 Sibiu, Romania; ateodoru77@yahoo.com

**Keywords:** propolis, chemical composition, HPLC, rutin, quercetin, molecular descriptors, antioxidant activity, antimicrobial activity, pathogenic bacteria

## Abstract

The impact of globalization on beekeeping brings new economic, scientific, ecological and social dimensions to this field The present study aimed to evaluate the chemical compositions of eight propolis extracts from Romania, and their antioxidant action and antimicrobial activity against seven species of bacteria, including pathogenic ones: *Staphylococcus aureus*, *Bacillus cereus*, *Bacillus subtilis*, *Pseudomonas aeruginosa*, *Escherichia coli*, *Listeria monocytogenes* and *Salmonella enterica* serovar Typhimurium. The phenolic compounds, flavonoids and antioxidant activity of propolis extracts were quantified; the presence of flavones and aromatic acids was determined. Quercetin and rutin were identified by HPLC analysis and characterized using molecular descriptors. All propolis samples exhibited antibacterial effects, especially against *P. aeruginosa* and *L. monocytogenes*. A two-way analysis of variance was used to evaluate correlations among the diameters of the inhibition zones, the bacteria used and propolis extracts used. Statistical analysis demonstrated that the diameter of the inhibition zone was influenced by the strain type, but no association between the propolis origin and the microbial activity was found.

## 1. Introduction

Spontaneous flora, crops and food production require pollinating insects, especially honey bees (*Apis mellifera*) [1]. The role of bees as pollinators makes them vital for agriculture, especially in maintaining plant biodiversity [2]. Anthropogenic activity, climate change, environmental pollution, intensive agricultural management and pesticide use have significantly reduced the number of pollinators [3,4,5].

Bee products are important sources of therapeutic agents [6]. A high percentage of all the available therapeutic drugs are derived from natural plants, microbes or animal compounds [7]. Terrestrial and aquatic species of plants and microorganisms produce unique bioactive substances that can help in the development of standardized phytomedicines with proof of quality, safety and efficacy [4,5,7,8,9,10,11].

Bee honey and propolis have been used since antiquity due to their health properties. Numerous studies conducted in recent years in various parts of the world highlighted the health benefits of these natural bee products [6,12,13,14,15,16].

Propolis is a substance produced from various plant sources, containing several polyphenolic constituents (mainly flavonoids and phenolic acids) [17]. The most investigated health benefits of the propolis include antimicrobial, immunomodulatory and cardio-protective properties; support for optimal neural function; and protection against cancer, the mechanism of action of which probably related to its antioxidant and anti-inflammatory activities [18].

The role of propolis in immune defense and its antioxidant properties derive from its bioactive phytochemical constituents. Compounds identified in propolis include phenolic acids, flavonoids, esters, diterpenes, sesquiterpenes, lignans, aromatic aldehydes, alcohols, amino acids, fatty acids, vitamins and minerals [19]. By considering the variety of compounds and phytonutrients, the great diversity of the observed biological properties can be explained [20,21]. The chemical compounds identified in propolis differ depending on the vegetation in the area of sampling and the harvesting period [22]. It was observed that the propolis from regions with a temperate climate (e.g., Europe and Asia) contains mainly simple phenolic acids, whereas in warmer climates (e.g., Brazil), lignans predominate [23]. Pinobanksin and pinocembrin accounted for 70% of the flavonoid content in a New Zealand propolis [24], a much higher percentage than in other regions, such as Brazil, Uruguay and China [25]. The antioxidant activity of propolis and its constituents is well documented; most studies have demonstrated reductions in various markers of oxidative stress [26,27,28]. The consumption of natural products containing phenolic compounds and flavonoids was found to be associated with lower incidences of cardiovascular diseases, cancer, diabetes and neurodegenerative diseases [29]. These substances, secondary plant metabolites, have at least one hydroxyl group attached to their aromatic rings, making them good electron donors, which explains their antioxidant activity [30,31]. Additionally, propolis’s antioxidant and immunomodulatory properties were found to interfere with SARS-CoV-2 metabolism [32].

All types of propolis, regardless of origin, present antimicrobial activity, indicating that this property is influenced by their composition as a whole rather than by individual compounds [27]. Globally, propolis is delivered through supplements or food and beverage additive systems [18], but the chances of an empirical remedy being approved in a drug or dietary supplement depend on chemical, technological, toxicological, preclinical and clinical tests.

To determine the biological activities of propolis, ethanolic or aqueous extracts need to be prepared following specific procedures [33]. Although standardized extracts, tinctures and mixtures of natural active ingredients currently exist, standardized (non-toxic) propolis extracts can cause side effects when overdosed on [34].

The present study aimed to evaluate the chemical compositions of eight propolis extracts of Transylvanian origin, and their antioxidant action and antimicrobial activities against seven species of bacteria, including pathogenic bacteria: *Staphylococcus aureus*, *Bacillus cereus*, *Bacillus subtilis*, *Pseudomonas aeruginosa*, *Escherichia coli*, *Listeria monocytogenes* and *Salmonella enterica* serovar Typhimurium. This research also analyzes possible correlations between the origin of the propolis and its antimicrobial activity, and between the type of pathogen and the diameter of the inhibition zone.

## 2. Materials and Methods

### 2.1. Propolis Sampling

Brown propolis samples (S1–S8) were collected in June and July 2019 from different areas of Transylvania, Romania: hilly (Caraș-Severin, Cluj, Mureș), plains (Arad, Timiș), mountainous (Alba) and sub-mountainous (Sibiu, Hunedoara) areas (Figure 1). All samples were tested in terms of chemical composition and antimicrobial properties against some microbial strains.

The honey bees belonged to the common *Apis mellifera* species. They were taken care of by local beekeepers in subsistence hives.

### 2.2. Chemical Characterization of Raw Propolis

#### 2.2.1. Qualitative Identification of Flavones’ Presence 

The protocol used for identifying the presence of flavones [35] was detailed in our previous study [36]. Fine propolis powder (5 g) was homogenized with ethanol 96% (20 mL), and the mixture was left to rest for 3 h, under periodical stirring. The extract was filtered and heated to evaporate the alcohol and increase its viscosity. After cooling, borax (5 g) was added and mixed well; then distilled water (10 mL) was added drop by drop and vigorously homogenized. After a few minutes at rest, a cloudy liquid was separated and filtered. A few drops were pipetted onto a strip of filter paper, and yellow spots were obtained. The presence of flavones was confirmed by color shifts observed on the yellow spots: reddish brown when adding uranyl nitrate crystals, gray when adding crystals of ferric sulfate.

#### 2.2.2. Identification of Aromatic Acids

The identification of aromatic acids [35] was conducted as described in our previous study [36]: Five milliliters of the solution prepared to identify the flavones (Section 2.2.1.) was precipitated with diluted sulfuric acid (1:10); then peroxide-free ethyl ether (10 mL) was added under vigorous stirring for one minute and the mixture was rested to allow the ether/aqueous layers to separate. The upper layer (extraction ether) was separated and collected in a glass beaker. Another 10 mL of ethyl ether was added to the remaining liquid, and the previous operation was repeated, the ether extract being collected in the same beaker. The ether solution was dehydrated by filtration on anhydrous sodium sulfate, then evaporated to dryness. The addition of 2n NaOH and KMnO_4_ drops generated under gentle heating specific smells of bitter almonds (benzoic aldehyde) and cinnamon (cinnamic aldehyde).

#### 2.2.3. Quantification of the Phenolic Compounds

This determination was conducted using the Folin-Ciocalteu method, as described in our previous study [36,37,38]. Raw ground propolis and ethanol were homogenized, filtered and rested. An equivalent quantity of Folin-Ciocalteu reagent was added to the ethanolic propolis extract. Spectrophotometric measurements (on a Lambda 20—Perkin Elmer UV/VIS, Washington, DC, USA) were carried out at 765 nm with distilled water serving as a blank. Readings of the total phenolic concentrations were compared to a standard curve of gallic acid.

#### 2.2.4. Determination of Flavonoid Content

The flavonoid content (an important quality index) was determined using the aluminum chloride colorimetric method, as described in our previous study [36,37,39,40]. Ethanolic propolis extracts and quercetin ethanolic dilutions needed to elaborate the standard curve were prepared. The diluted standard quercetin solutions and the propolis extracts were separately mixed with 95% ethanol and aluminum chloride, then incubated at room temperature. The absorbance of the reaction mixtures was measured at 415 nm against a blank on a Lambda 20—Perkin Elmer UV/VIS spectrophotometer. The total flavonoid content was derived from the calibration plot and expressed as quercetin equivalents (mg EQ/g). All analyses were carried out in triplicate.

### 2.3. The Antioxidant Activity of Propolis

Raw propolis was macerated in a 70% ethanol solution (1:100 *w/v*) for 24 h at room temperature under continuous stirring, and then evaporated till dry. In order to measure the 2,2-diphenyl-1-picrylhydrazyl (DPPH) free radical-scavenging activity, a reaction mixture containing DPPH 0.1 mM ethanolic solution and 0.6 mg/mL propolis solution was then prepared. The decrease in the absorption value of the DPPH solution at λ = 515 nm was spectrophotometrically measured in a quartz cuvette (1 cm^3^). Absorbance (A) was recorded at the moment the reaction was initiated, then exactly 10 and 20 min later. The antioxidant activity commensurate with DPPH percentage was calculated using the formula: %RSA = (A_DPPH_ − A_sample_)/A_DPPH_ × 100 [41,42].

### 2.4. Preparation of the Propolis Extracts

#### 2.4.1. Preparation of the Aqueous Propolis Extracts

The aqueous propolis extracts were obtained as described in our previous study [36]. Propolis powder aqueous suspension was refluxed for one hour, 2× centrifuged-filtered, then maintained at water’s boiling point until 80% of the initial mixture was evaporated. The aqueous propolis extracts were stored in a chilly dry dark space; samples of 0.1 g/mL were used to test their antimicrobial activity.

#### 2.4.2. Preparation of the Ethanolic Extracts of Propolis

A mixture of 50 g of finely granulated and purified propolis and 250 mL of absolute ethanol was refluxed for one hour and twice centrifuged and filtered using a vacuum-connected filter. It was then centrifuged again at 10,000 rpm and filtered through a low porosity surface connected to a vacuum outlet, before being carefully concentrated up to 20%. The ethanolic propolis extracts were stored in conditions similar to those in Section 2.4.1. Three different concentrations were tested to determine the best extraction yields of quercetin and rutin: E1 (99%), E2 (50%) and E3 (25%).

### 2.5. Identification of Quercetin and Rutin in Aqueous and Ethanolic Propolis Extracts

A quantitative analysis of the flavonoids was conducted using a Hewlett Packard Agilent 1100 HPLC System with UV detection (Marshall Scientific, Hampton, NH, USA) equipped with a Nucleosil C18 column under the following conditions: stationary phase particle size of 5 μm; column dimensions 150 × 4.6 mm × mm; eluent: acetonitrile:water = 1:1; eluent flow rate: 1 mL/min; wavelength: 365 nm; temperature: 30 °C; injected volume: 20 μL. The aqueous and ethanolic propolis extract samples were dissolved in ethanol (5 mg/mL) and filtered through a 0.45 μm filter prior to injection into the HPLC system.

### 2.6. Molecular Descriptors for Rutin and Quercetin

The 2D structures of rutin and quercetin were built based on the basic structure of flavonoids and introduced into the HyperChem version 7.1 molecular modelling program. The most stable geometric arrangements (characterized by the lowest energy configuration) were determined by semi-empirical PM3, RHF molecular orbital calculations, in a vacuum, using the Polak–Ribière minimization algorithm with a root mean square (RMS) energy gradient of 0.01 Kcal/Å·mol [43].

The structural properties calculated for quercetin and rutin based on molecular modelling included the van der Waals surface (A), molar volume (V), the partition coefficient (LogP), refractivity (R), polarizability (α), the dipole moment (μ_t_), standard enthalpy of formation (H_formation_) and energy of hydration (E_hydr_). The number of phenolic hydroxyl groups and antioxidant activity data were taken from other studies.

### 2.7. Antimicrobial Activity of Aqueous Propolis Extracts

#### 2.7.1. Micro-Organisms and Culture Conditions

Seven bacterial species (both Gram-positive and Gram-negative) were used for testing the antibacterial properties of the aqueous propolis extracts. To assess the antibiotic susceptibility of the selected strains, the disk diffusion method was used according to CLSI-recommended procedures [44]. The antibiotic sensitivity of the strains was determined by measuring the diameters of the inhibition zones, using ciprofloxacin (Bio-Rad, Marnes-la-Coquette, France) as a positive control. The antimicrobial properties were tested only for aqueous propolis extracts, the alcoholic extracts (due to the presence of ethanol, with known antimicrobial activity) being susceptible to confounding the diameters of the inhibition zones.

The following bacterial strains (most of them pathogenic for humans) were used: *Staphylococcus aureus* (ATCC 25923), *Bacillus cereus* (ATCC 11788), *Bacillus subtilis* subsp. *spizizenii* (ATCC 6633), *Pseudomonas aeruginosa* (ATCC 27853), *Escherichia coli* (ATCC 25922), *Listeria monocytogenes* (ATCC 19115) and *Salmonella enterica* serovar Typhimurium (ATCC 14028), provided by MicroBioLogics Inc. (St. Cloud, MN, USA).

Direct colony suspensions of overnight cultures were diluted in nutrient broth (Mikrobiologie Labor-Technik, Arad, Romania) and adjusted to a 0.5 McFarland turbidity standard measured using a McFarland Densitometer (Mettler Toledo, Columbus, OH, USA). According to the manufacturer, the McFarland 0.5 standard corresponds approximately to a homogeneous suspension of 1.5 × 10^8^ CFU (colony forming units)/mL.

#### 2.7.2. Determination of the Antibacterial Properties of the Aqueous Propolis Extracts—Agar Disk Diffusion Method

Mueller–Hinton agar (Merck KGaA, Darmstadt, Germany) was used as culture medium. The depth of the agar was 4 mm (25 mL in 9-cm-in-diameter Petri dishes). The entire surface of the Petri dish was inoculated by stretching the suspension (1 mL broth culture of a bacterial strain) with a sterile cotton swab. After inoculation, the plates were kept for 15 min at 37 °C, to allow the inoculum to be absorbed in the agar.

Amounts of 50 µL of each propolis extract (at a concentration of 0.1 g/mL) were impregnated in ~6 mm filter paper disks prepared in the laboratory. Discs with 5 µg ciprofloxacin (Bio-Rad, France) were used as positive controls. The discs were sterilely deposited on the surfaces of the culture media, being applied at approximately the same distance from the edge of the plate and from each other. The Petri dishes were kept at 5 °C for 120 min, then incubated 24 h at 37 °C for bacterial growth. A transparent ruler placed on the back of the plate and a DIN 862 ABS digital caliper (Fuzhou Conic Industrial Co. Ltd., Fuzhou, China) with ±0.01 mm accuracy were used for measurements. The antimicrobial activity was evaluated by quantifying the resulting inhibition zones.

#### 2.7.3. Minimum Inhibitory Concentrations (MICs) of the Aqueous Propolis Extracts

The dilution method was used to determine the MICs. The aqueous propolis extracts were mixed with deionized water (*v/v*) to obtain final dilutions of 1/1, 1/4, 1/8, 1/16, 1/32, 1/64 and 1/128. The antimicrobial activity of the diluted propolis extracts was evaluated by the disc diffusion method, as described earlier (see Section 2.7.2). MIC was the lowest concentration where the inhibition of microbial growth was observed.

### 2.8. Statistical Analysis

Two-way (ANOVA) tests were used to evaluate the connections between the diameter of the inhibition areas for various pathogenic strains and the Transylvanian propolis samples [45]. The significance level was set to α = 0.05. The analysis of variance was performed using an Origin 8.0 (OriginLab Corp., Northampton, MA, USA) software. A Pearson’s correlation was used to establish if the content of phenols and flavonoids correlated with the MIC for each microbial strain.

## 3. Results

### 3.1. Chemical Characterization of the Propolis Samples

The chemical parameters of the propolis samples (the content of phenolic compounds and flavonoids, the presence of flavones and aromatic acids) are presented in Table 1. All tests were carried out in triplicate assays.

All samples indicated the presence of bioactive compounds. As seen in Table 2, flavones were identified in all eight samples, and aromatic acids were absent in sample S3. All propolis samples presented significant free radical scavenging activity (RSA), ranging between 10.29% and 19.31%.

### 3.2. Identification of Quercetin and Rutin in Aqueous and Ethanol Propolis Extracts

The efficiency of ethanol for quercetin and rutin extraction yields was tested on sample S4, harvested from a mountainous area of Alba County. Ethanolic extracts of 99%, 50% and 25% were used, along with aqueous extracts. Five consecutive replicates were performed for each extract, and the relative standard deviations (RSD) were calculated.

When quantifying the main two flavonoids in S4, quercetin extraction yields were found to be highest in the 99% ethanolic extracts; for rutin the opposite was observed. Based on these results, it was concluded that 99% ethanol and aqueous extracts should be considered for further investigations (S1–S3, S5–S8).

Table 2 highlights the quercetin and rutin concentrations (mg/mL) in the eight propolis extracts.

**Table 2 ijerph-19-02640-t002:** Quercetin and rutin concentrations in the propolis extracts.

Sample	Propolis Extract	Quercetin (mg/mL); RSD%		Rutin (mg/mL); RSD%	
S4	Aqueous	0.74; 2.94		0.0143; 1.47	
	Ethanolic 25%	0.69; 2.83		0.0120; 1.38	
	Ethanolic 50%	0.83; 1.91		0.0057; 2.67	
	Ethanolic 99%	1.12; 1.64		0.0048; 2.81	
		**Ethanolic (99%)**	**Aqueous**	**Ethanolic (99%)**	**Aqueous**
S1		1.04	0.57	0.0030	0.0153
S2		1.02	0.62	0.0127	0.0168
S3		1.20	0.62	0.0094	0.0080
S5		0.92	0.67	0.0027	0.0148
S6		1.10	0.83	0.0071	0.0196
S7		0.86	0.64	0.0046	0.0128
S8		0.83	0.82	0.0018	0.0171

The high-performance liquid chromatography (HPLC) analysis of the propolis samples collected from Transylvania revealed the presence of two flavonoids: quercetin and rutin. While quercetin exhibited higher concentrations in ethanolic extracts, ranging from 0.83 to 1.20 mg/mL, compared to 0.57–0.83 mg/mL in aqueous extracts, rutin was less present in ethanolic extracts (0.0018 to 0.0127 mg/mL vs. 0.0080 to 0.0196 mg/mL in aqueous extracts; see Table 2).

### 3.3. Molecular Descriptors for Rutin and Quercetin

The two flavonoid compounds rutin and quercetin were studied in terms of structural characteristics using computational chemistry. The modelling of the molecular structures was carried out in a vacuum by semi-empirical methods (AM1, RHF). The structures were optimized using the Polack-Ribiere algorithm with an RMS to a maximum energy gradient of 0.01 kcal/Å·mol. The system’s energy and other molecular properties were determined using the Schrödinger equation.

Table 3 contains data corresponding to the calculated properties for the structures of the two analyzed flavonoids.

### 3.4. Antimicrobial Activity of Aqueous Propolis Extracts

The antibacterial effects of the analyzed aqueous propolis extracts, compared to the effect of ciprofloxacin, used as positive control, are presented in Table 4. As can be seen, all samples presented some antimicrobial activity against the bacterial strains tested.

The inhibition areas were between 16 and 32 mm in diameter; in some cases they were even larger than those produced by ciprofloxacin. The largest was observed for S4 (mean diameter 28.85 mm), and the weakest antibacterial effect was observed for S5 (mean diameter 24.14 mm). The species most sensitive to the antibacterial activity of propolis were *P. aeruginosa* and *L. monocytogenes*, both with a mean inhibition zone diameter (30.25 mm) larger than the one produced by ciprofloxacin.

#### Minimum Inhibitory Concentration (MIC)

The 1/1, 1/4 and 1/8 dilutions were found to manifest an inhibitory effect on all species, except S5 (the 1/8 dilution, for *S. aureus* and *E. coli*). Greater dilutions (1/16, 1/32, 1/64) had partial effects on certain bacterial species. MIC values are presented in Table 5.

### 3.5. Statistical Analysis

The bifactorial analysis of variance tested the simultaneous interaction of two independent variables: the Transylvanian aqueous propolis extracts and the diameter of the inhibition area for the studied strains. The computation of variances caused by each independent parameter, including residual dispersion caused by accidental factors, produced the following results: S_1_ = 41,820, S_2_ = 40,873.4, S_3_ = 41,038.2 and S_4_ = 40,716.1. The variance in the diameter of inhibition zones between propolis extracts was s_1_^2^ = 22.47; between strains it was s_2_^2^ = 53.69; and residual, s_r_^2^ = 22.35.

As seven degrees of freedom were found for the propolis extracts (ν_1_) and six for the strains tested (ν_2_), since F_col_ = 1.00 < F_0.05_ = 2.24, the null hypothesis that the mean values of the columns are equal was accepted. It was concluded that there are insufficient proofs to confirm that the propolis extracts influenced the diameters of the inhibition areas. Additionally, because F_row_ = 2.40 > F_0.05_ = 2.32, the hypothesis that mean values of the rows are equal was rejected, and it was concluded that the type of bacteria affected the inhibition areas. The significance level was α = 0.05.

The two-way ANOVA analysis for antibacterial activity found no correlations between the origins of the propolis extracts and the diameters of the inhibition areas, but the type of pathogenic strain was found to influence the diameter of the inhibition area.

Pearson correlation analysis between the MIC values and flavonoids content demonstrated a weak and moderate negative linear relationship (*p* < 0.050). The Pearson’s correlation coefficient was between −0.651 and −0.233. The correlations between MIC and phenols content were also low, suggesting no linear relationship (R^2^ range between −0.162 and +0.153).

## 4. Discussion

Many researchers have focused on the possible use of honeybee products for the treatment of the symptoms of disease. The flavonoids from propolis and honey (rutin, naringin, caffeic acid phenyl ester, luteolin) may inhibit viral spike fusion in host cells, viral–host interactions that trigger cytokine storms and viral replication [47]. Flavonoids from ethanolic propolis extracts have high binding activity, thereby blocking the angiotensin-converting enzyme 2 (ACE-2) receptors [48]. Quercetin is used in the treatment of SARS-CoV-2, and its action has an inhibitory influence on the viral polymerase, as found in other RNA viruses [49]. Rutin has also been identified as a potential antiviral drug inhibiting the SARS-CoV-2 main protease (M^pro^) [50].

Due to the large variations in composition seen in the propolis specimens found worldwide, each new sample should be analyzed and classified according to its chemical profile. Due to the diversity of flora used by bees to produce propolis, specific to each geographical region, the number of constituents that can be identified in propolis is in the hundreds [51,52,53]. Chemical analyses were performed for the compounds likely to influence the propolis extracts’ antioxidant and antibacterial activity, i.e., phenolic compounds, flavonoids, flavones and aromatic acids. Higher concentrations were observed in sample S4 from a mountainous area in Alba County.

The effects of ethanol concentration on quercetin and rutin extraction were tested in sample S4. Although methanol has higher ionic strength, which is important for effective extraction of bioactive compounds, ethanol and water are the solvents of choice, as they are considered “green” solvents [54]. Aqueous and ethanolic extracts of 99%, 50% and 25% were used, the findings being that quercetin extraction yields were highest in the 99% ethanolic extracts, and that for rutin the best results were in aqueous extracts.

Several analytical methods have been developed for the identification and isolation of chemical compounds such as polyphenols and flavonoids. HPLC is a technique commonly used to analyze flavonoids [55,56,57]. Liquid chromatography–mass spectrometry (LC-MS) has also been employed for the identification of phenolic compounds [58]. Yang et al. [59] and García-Viguera [60] compared the usual HPLC-DAD method with the more powerful gas chromatography–mass spectrometry (GC-MS) technique for the identification of rutin, quercetin, luteolin, genistein, galangin and curcumin in propolis. Other HPLC analyses revealed that the phenolic content of propolis extracts generally contains significant quantities of crysin, galangin, pinostrobin, pinobanksin and pinocembrin [61,62].

Propolis can be classified into three major classes, based on its predominant color: green, red or brown, depending on the geographical origin [52,62,63]. The antimicrobial properties vary depending on the composition; the antimicrobial effects of distinct types of propolis differ even when tested on identical bacterial strains. The geographical origin is also known to influence the mechanism of action. The green propolis is particularly indicated for wound healing [64]. The samples used in this study came in pale, medium and dark shades of brown. The high concentrations of phenolic and flavonoid compounds are probably specific to this type of Transylvanian propolis, explaining the antibacterial effect observed in all the samples.

The cause most likely to influence the physiological effects (e.g., antiallergic, antiviral, anaesthetic, antitumor, anti-inflammatory actions) observed as a result of flavonoid consumption is the modification of the enzymatic activity through phosphorylation, antioxidant action or gene repression [65]. The mechanisms by which these propolis compounds inhibit the activity and development of bacteria are the prevention of bacterial cell division and the induction of dysfunctions in their cytoplasm [18]. Flavonoids were shown to inhibit nucleic acid synthesis, energy metabolism of bacteria and their biofilm formation [66], proving to be effective anti-bacterial agents.

Due to the great diversity of the propolis subgroups, large variations in their flavonoid content are being observed [67]. The pharmacological activity of flavonoids is mainly due to their structural characteristics as tricyclic compounds presenting reactive free radicals [19].

Findings on the structural characteristics impacting the activity of certain compounds determined through molecular modelling can be extended to others with similar structures. The number and position of hydroxyl groups were found to greatly influence the flavonoids’ antioxidant activity. The free radical scavenging mechanism leads to the formation of less reactive phenoxy radicals. The smaller the number of hydroxyl groups, the greater the scavenging ability of a compound. Careful interpretation of the molecular descriptors provides a fresh perspective on the antioxidant activity of flavonoids based on their structural characteristics [68,69].

The antioxidant activity of different propolis samples also depends on their total phenolic and flavonoid concentrations. The lowest free radical-scavenging activity (RSA) was observed for S2, a sample collected from the Arad County plain area, and the highest was observed for S4. Figure 2 illustrates the antioxidant activity vs. flavonoid content in the examined propolis samples.

The RSA of 2,2-diphenyl-1-picrylhydrazyl (DPPH) determined through spectrophotometric analysis in the examined propolis samples correlated with data obtained using the total flavonoid content (R^2^ = 0.73). Other studies found R^2^ values of 0.5 or 0.76 [70,71].

Numerous studies have found that various propolis types have significant antimicrobial activity against a wide range of pathogens: *Bifidobacterium infantis*, *Enterococcus faecalis*, *E. coli*, *Helicobacter pylori*, *L. monocytogenes*, *Neisseria gonorrhoeae*, *S. aureus*, *Staphylococcus epidermides*, *Streptococcus pyogenes*, *B. cereus*, *P. aeruginosa*, *Enterococcus faecium* or *Candida albicans* [13,23,36,72,73,74,75]. Some of these microorganisms were isolated from patients and exhibited resistance to classes of antibiotics.

The microbiological activity was reflected in the examined propolis samples by the relative concentrations of quercetin and rutin in the extracts used, as determined when separating them at retention times of 3.1 (keto-form quercetin), 3.4 min (enol-form quercetin) and 2.1 min (rutin).

Differences in extraction methods and solvents used may cause some variability in the biological activities of propolis [76]. It is known that raw propolis can only be used when purified following solvent extraction, mainly in ethanol or methanol [77].

The solvent’s nature also influences the nutrient profile [61]. Certain differences in the concentrations of quercetin and rutin were observed when comparing aqueous and ethanolic extracts. While quercetin concentrations were higher in the ethanol extracts, in most propolis samples, rutin concentrations were higher in the aqueous extracts. To avoid possible bacterial inhibition of the ethanolic extracts, only the aqueous propolis extracts were used when determining the antibacterial properties of the samples. Most studies have been conducted on ethanolic propolis extracts; there are less data on the biological activity of aqueous propolis extracts.

In this study, both Gram-negative and Gram-positive bacteria were used. Antibacterial effects were observed for all aqueous propolis extracts (0.1 g/mL concentration) against all tested bacteria, occasionally outshining the antibiotic used as the control. The antibacterial effects were observed even at very low concentrations—samples S4 and S6 being active at 1/64 dilutions against some strains. Although similar results were reported in another study on red propolis [78], MIC values were slightly higher in this case, perhaps because of using aqueous propolis extracts in contrast to ethanolic ones as in most other studies.

The MIC values reflected the significant inhibitory effects induced by propolis on the growth of the tested microorganisms. The analyzed propolis samples presented antimicrobial activity against all bacterial species. Low correlations between MIC and flavonoids indicated a negative linear relationship, and low correlations between MIC and the phenolic compounds suggested a slightly positive linear relationship, which are similar the findings of a previous study on honey [79].

The antimicrobial effect of propolis differed from one bacterial species to another. The species more sensitive to the action of propolis were *P. aeruginosa* and *L. monocytogenes*, these two being less inhibited by ciprofloxacin. Differences in the antimicrobial activity of these propolis samples can be explained by variations in their chemical compositions depending on their areas of origin. The antibacterial effect being observed for all Transylvanian samples can be attributed to the high content of polyphenols and flavonoids. Quercetin is known for its strong antimicrobial activity.

The polyphenolic content of various propolis samples varied from 143 to 324 mg GAE/g in a previous study [63]. As can be seen in Table 1, in the analyzed samples, phenolic compounds varied from 134.7 to 203.3 mg GAE/g, these values probably being specific to the type of propolis produced in the respective area.

Samples S1, S4, S6 and S7 presented average diameters of their inhibition zones of over 28 cm (Table 4), indicating good antibacterial effects. A correlation between the antimicrobial effects of these samples and their chemical compositions can be observed. Samples S1 (187.9 mg GAE/g), S4 (203.3 mg GAE/g) and S7 (190.6 mg GAE/g) had the highest content of phenolic compounds; and the highest flavonoid content was found in samples S1 (83.60 mg QE/g), S3 (86. 48 mg QE/g), S4 (90.54 mg QE/g) and S7 (80.19 mg QE/g). Flavones and aromatic acids were identified in all these samples.

Higher quercetin concentrations in the ethanol extracts were observed in samples S3 (1.20 mg/mL), S4 (1.12 mg/mL) and S6 (1.10 mg/mL). In the aqueous extracts the highest concentrations were found in samples S4 (0.74 mg/mL), S6 (0.83 mg/mL) and S8 (0.82 mg/mL). In the aqueous extracts, the highest rutin concentrations were found in samples S2 (0.0168 mg/mL), S6 (0.0196 mg/mL) and S8 (0.0171 mg/mL). For ethanolic extracts, these were samples S2 (0.0127 mg/mL), S3 (0.0094 mg/mL) and S6 (0.0071 mg/mL). Samples S2, S3, S4, S6 and S8 also had the highest total flavonoid content.

When comparing these samples in regard to the diameters of the inhibition zones against the microbial strains, S4 and S6 were the most effective, but the others caused appreciable diameters as well. Sample S4 (originating from Alba County) presented the most intense antibacterial activity, followed by S6 (Hunedoara County) and S7 (Cluj County). One can notice that sample S4 also had the highest concentrations of phenolic compounds and flavonoids (see Table 1).

The ANOVA tests for antimicrobial activity found no correlations with the propolis extracts’ origins, but the selected strains influenced the diameters of the inhibition zones, confirming that the antibacterial activity of the various aqueous extracts of propolis differed with the bacterial strains tested.

Regarding Pearson’s correlation coefficients between MIC and flavonoid and phenol content for the studied microbial strains, the correlations between MIC and flavonoids content were low, indicating a negative linear relationship between the two variables. The correlations between MIC and phenol content were also low, suggesting a positive linear relationship. This study focused on propolis as a natural antibiotic and as a possible alternative drug. Its results highlight the potential use of propolis extracts for combating bacterial infections. The structural and functional characteristics of various types of propolis will determine their clinical usefulness. Further studies on Transylvanian propolis are needed to determine other chemical compounds with potential antibacterial effects and to standardize propolis extracts in terms of activity and composition, to guarantee their quality and safety of use.

## 5. Conclusions

Propolis samples of different origins from Transylvania, Romania was evaluated for determination of phenolic compounds and flavonoids. The propolis sample from Alba County exhibited the highest content of bioactive compounds. Propolis flavonoids such as quercetin and rutin are two of the components responsible for the antimicrobial and antiviral activity that were quantified in both ethanolic and aqueous extracts. Differences in the concentrations were observed: while rutin was found in a higher concentration in the aqueous propolis extract, the quercetin concentration in the aqueous propolis extract was about half of that in the ethanolic extract.

Only aqueous propolis extracts were used when determining the antibacterial properties of the samples. Most studies have been conducted on propolis ethanolic extracts, so les data on the biological activity of propolis aqueous extracts are available. Correlations between the chemical compositions of the propolis samples and their antioxidant and antimicrobial activities were observed.

When comparing the quercetin and rutin concentrations determined in aqueous propolis extracts to the total content of flavonoids, and to the diameters of the inhibition zones of the microbial strains, it was observed that the samples from the mountainous area of Alba County and sub-mountainous area of Hunedoara contained the highest concentrations of bioactive compounds and exhibited the best antimicrobial activity.

Two-way analysis of variance found no correlations between the origins of the Transylvanian propolis and their antibacterial activity, but the diameter of the inhibition zone was found to be influenced by the species of pathogen. The antibacterial effects of the propolis extracts against *P. aeruginosa* and *L. monocytogenes* strains in particular, argue for their potential use in alternative medical practices.

## Figures and Tables

**Figure 1 ijerph-19-02640-f001:**
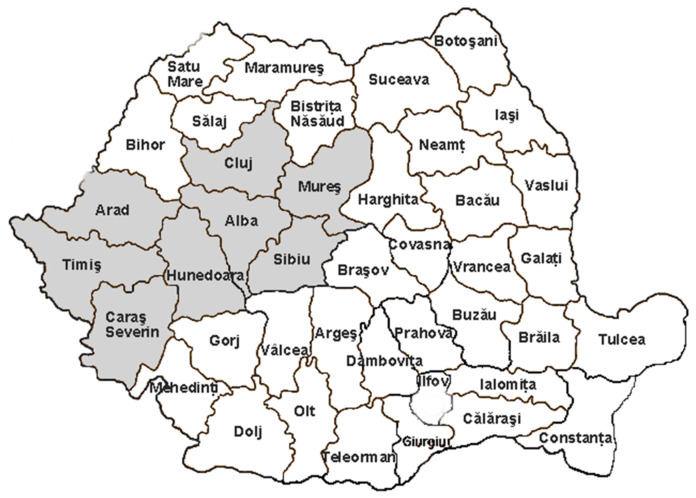
Map of Romania highlighting the Transylvanian propolis sampling counties.

**Figure 2 ijerph-19-02640-f002:**
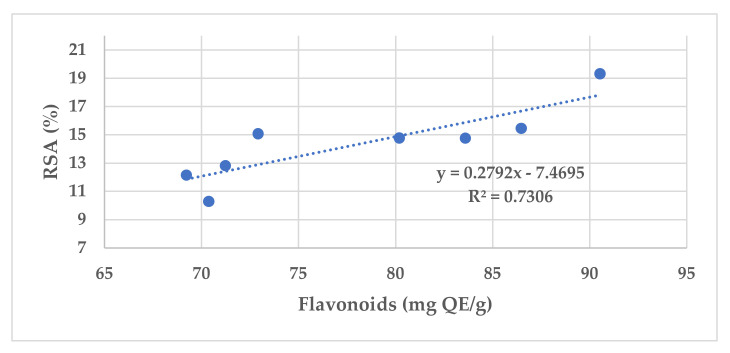
Correlation of flavonoid content and antioxidative activity in the examined propolis samples.

**Table 1 ijerph-19-02640-t001:** The chemical parameters of the propolis samples.

Sample	Phenolic Compounds (mg GAE/g)	Flavonoids (mg QE/g)	Flavones	Aromatic Acids	RSA (%)
S1	187.9 ± 6.25	83.60 ± 0.05	+	+	14.75
S2	172.2 ± 6.14	70.37 ± 0.03	+	+	10.29
S3	158.8 ± 5.27	86.48 ± 0.02	+	−	15.46
S4	203.3 ± 7.28	90.54 ± 0.06	+	+	19.31
S5	181.5 ± 6.10	72.92 ± 0.07	+	+	16.07
S6	134.7 ± 4.09	71.24 ± 0.02	+	+	13.82
S7	190.6 ± 5.26	80.19 ± 0.01	+	+	14.78
S8	169.1 ± 8.39	69.23 ± 0.04	+	+	11.15

GAE—gallic acid equivalents; QE—quercetin equivalents; RSA—radical-scavenging activity; “+”—identified compounds; “−”—unidentified compounds.

**Table 3 ijerph-19-02640-t003:** Calculated molecular descriptors for rutin and quercetin.

Molecular Descriptor	Rutin	Quercetin
A [Å^2^]	783.58	443.05
V [Å^3^]	1320.04	704.27
Log P	11.21	2.52
R [Å^3^]	97.01	77.12
α [Å^3^]	44.18	23.99
H_formation_ [kcal/mol]	−598.683	150.6105
E_hidr_ [kcal/mol]	−22.54	−10
μ_t_ [D]	2.162	2.352
No. of –OH phenolic groups	4	5
Antioxidant activity [%] *	90.9 *	89.9 *

* Experimental antioxidant activity data—from Burda and Oleszek [46].

**Table 4 ijerph-19-02640-t004:** Disk susceptibility assays of aqueous propolis extracts and ciprofloxacin against microbial strains.

Strain	Inhibition Diameter Area (mm)										
	Sample No. (0.1 g/mL)								Total ∑x_j_	Average x_j_	Ciprofloxacin (5 µg)
	S1	S2	S3	S4	S5	S6	S7	S8			
*S. aureus*	16	19	31	30	19	25	32	18	190	23.75	30
*B. cereus*	28	29	27	25	26	27	28	29	219	27.37	30
*B. subtilis*	29	23	27	28	24	29	27	29	216	27	30
*P. aeruginosa*	32	31	31	27	32	30	29	30	242	30.25	25
*E. coli*	32	26	19	32	18	30	27	22	206	25.75	29
*L. monocytogenes*	30	30	31	29	30	31	30	31	242	30.25	24
*S. typhimurium*	30	21	18	31	20	29	26	20	195	24.37	29
Total ∑x_i_	197	179	184	202	169	201	199	179	∑x_ij_ = 1510		-
Average x_i_	28.14	25.57	26.28	28.85	24.14	28.71	28.42	25.57			

**Table 5 ijerph-19-02640-t005:** The minimum inhibitory concentrations of the aqueous propolis extracts.

Sample No.	MIC (mg/mL)						
	*S. aureus*	*B. cereus*	*B. subtilis*	*P. aeruginosa*	*E. coli*	*L. monocytogenes*	*S.* *typhimurium*
S1	12.5	6.25	6.25	3.12	6.25	3.12	6.25
S2	6.25	12.5	6.25	6.25	6.25	6.25	6.25
S3	6.25	3.12	6.25	3.12	6.25	3.12	12.5
S4	3.12	1.56	3.12	1.56	3.12	1.56	3.12
S5	25.0	6.25	12.5	12.5	25.0	6.25	12.5
S6	6.25	3.12	3.12	1.56	3.12	1.56	3.12
S7	6.25	3.12	3.12	3.12	6.25	3.12	3.12
S8	12.5	6.25	6.25	6.25	6.25	6.25	12.5

## Data Availability

Data are contained within the article.

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
