# Peer review of "Potential Effects of Romanian Propolis Extracts against Pathogen Strains"

_ijerph, 2022, doi:10.3390/ijerph19052640_

Round 1
Reviewer 1 Report
Comments and suggestions can be found in the attached file.

Author Response
Dear Reviewer,
Thank you for your appreciations and recommendations. The text has been improved in several sections, according to your recommendations. In attachment are our responses to your comments.

Reviewer 2 Report
All Comments and Suggestions the Authors will find in the Word document

Author Response

(The authors gave the same response as above.)

Reviewer 3 Report
The manuscript is titled “Beekeeping during the COVID-19 Pandemic and the Potential Effect of Romanian Propolis Extracts against Pathogen. The author’s research study explores the impact effects of COVID-19 on the beekeeping. Additionally, the authors investigate the chemical composition of propolis samples from Romania and their antibacterial properties against seven bacterial strains. The author’s investigation of the chemical composition of propolis samples and their antibacterial properties is a well-designed and informative study. The addition of “reviewing the impact of COVID-19 on the beekeeping industry” is out of place in this manuscript. There is no discussion of this in the Introduction, Methods or Results. There is one line in the Discussion that mentions this topic. All aspects of this manuscript need to be rewritten and the removal of “Beekeeping during the COVID-19 Pandemic” from the title and Abstract. This study was not performed and does not belong here. There are too many tables, and two tables labeled Table 8. Figure 2 is missing, Figures 3 and 4 are unnecessary. Additional comments are made in the PDF.

Author Response

(The authors gave the same response as above.)

Round 2
Reviewer 3 Report
The manuscript is titled “Potential Effects of Romanian Propolis Extracts against Pathogen Strains” The author’s research study evaluated the chemical composition of eight propolis extracts from honey bee hives in Romania and their antioxidant action and antimicrobial activity against 7 bacterial strains. The sample size used was adequate for the experimental design. The data presented is a bit difficult to comprehend with the use of many tables. Some of the tables can be combined or eliminated and the information discussed in the body of the manuscript. The authors did an excellent job of eliminating the COVID study; this made for a much better manuscript. The manuscript and the research performed has merit and is described well with the edits made. I feel the manuscript is now well prepared, and that this study can be replicated with the information provided. Additionally, I feel that proper methods were used to justify the conclusion. I recommend the manuscript for publication with some minor edits that I have made in the PDF.

Author Response
Dear Reviewer,
Thank you for your appreciations and recommendations.
The text has been improved in all the sections, according to the recommendations.
Attached are our responses to your comments.
